# Nitrogen Self-Doping Carbon Derived from Functionalized Poly(Vinylidene Fluoride) (PVDF) for Supercapacitor and Adsorption Application

**DOI:** 10.3390/mi13101747

**Published:** 2022-10-15

**Authors:** Yantao Zheng, Qifei Liu, Xingyu Guan, Yuan Liu, Shengqiang Nie, Yi Wang

**Affiliations:** 1College of Chemistry and Material Engineering, Guiyang University, Guiyang 550005, China; 2Xifeng Phosphorite Mine Co., Ltd., Guiyang 551100, China; 3College of Material Engineering, Saint Petersburg State Technical University, 190013 Saint Petersburg, Russia

**Keywords:** supercapacitor, carbon, different content nitrogen self-doping, adsorption

## Abstract

A new synthetic strategy has been developed for the facile fabrication of a N-doped porous carbon (NC-800) material via a facile carbonization of functionalized poly(vinylidene fluoride) (PVDF). The prepared NC-800 exhibits good specific capacitance of 205 F/g at 1 A/g and cycle stability (95.2% retention after 5000 cycles at 1 A/g). The adsorption capacity of NC-800 on methylene blue and methyl orange was 780 mg/g and 800 mg/g, respectively. The facile and economical method and good performance (supercapacitor and adsorption) suggest that the NC-800 is a promising material for energy storage and adsorption.

## 1. Introduction

Energy crises and serious environmental concerns have accelerated the development of high-energy devices that are both efficient and environmentally friendly [1,2,3]. Supercapacitors, as a novel energy storage device, have a higher power density, increased charge and discharge rates, and a longer cycle life [4,5,6]. There are two types of supercapacitors: electrical double-layer capacitors (EDLCs) and pseudocapacitors [7,8,9]. The EDLC has outstanding cycle stability, and a wide voltage window but low specific capacitance. A cost-effective method of enhancing electrochemical properties is to develop superior electrode materials. The EDLCs require a material with large specific surface area, low resistance, and rapid electron transport [10,11,12,13,14,15].

Adsorption properties are regarded as the most successful technology for porous materials due to their simplicity of operation and cost-effectiveness. For most synthetic dyes, removal of effluents from water is an urgent need, since they can cause severe adverse effects on human health and ecosystems when discharged into water. Recently, several approaches for dye removal from wastewaters have been developed, which include ion exchange, adsorption, and ultrafiltration. Due to their superior chemical stability and adsorption speed, porous carbon materials are considered ideal. They exhibit an excellent adsorption capacity for a variety of dyes [16,17,18,19,20].

Carbon has been widely used for electrode materials for supercapacitors due to moderate cost and superior electrochemical stability [21,22]. Moreover, carbon cannot offer higher capacitance due to the poor conductivity. Recent studies focused on improving the electrochemical properties of carbon-based supercapacitors by introduction of heteroatoms. Heteroatom doping significantly improve the surface wettability and the number of electrochemical active sites [23,24,25,26]. Nitrogen atoms are similar to carbon atoms in structure. Introducing nitrogen atoms in carbon materials can change the surface properties and the hydrophilicity of the materials, enhance the conductivity of the materials, and increase the electrochemical reaction sites. When nitrogen atoms are incorporated into carbon materials, bends and dislocations will occur in the carbon layer. As these defects have asymmetric electronic structures, the pore structure of the material surface is adjusted. The doping of N element can improve the chemical properties of the carbon material surface. The nitrogen atoms mainly exist in three forms on the carbon material skeleton, namely graphite nitrogen, pyrrole nitrogen, and pyridine nitrogen. Among them, graphite nitrogen can increase the conductivity of the material. The nitrogen atoms of pyridine nitrogen and pyrrole nitrogen are prone to oxidation–reduction reactions, which can provide a large number of pseudocapacitances to improve the capacitance performance of materials. Nitrogen doping can not only greatly improve the specific capacitance of the material, but also rapidly charge and discharge at high current density [27,28]. PVDF has attracted extensive attention due to low cost, high mechanical strength, and three-dimensional interconnected network structure [28]. Therefore, it is an excellent carbon material precursor. Currently, the common methods employed for modification of polymeric material include grafting, coating, and blending. As a type of bleeding method, in situ cross-linked polymerization is more convenient due to the involvement of a one-pot procedure. In an in situ cross-linked polymerization process, polymerization and cross-linking occur simultaneously; thus, it is a convenient and simple approach for large-scale fabrication at industrial scale [29] and has been widely used to design functional materials in the fields of fuel cells, water treatment, stimuli-responsive materials, ion-exchange membranes, and toxin removal [30,31,32,33]. As the molecular chains of the polymeric matrix and polymeric additive become entangled, the N-containing additive of N-vinyl pyrrolidone (NVP) can become “locked” into the polymeric matrix, thus effectively preventing wash out of hydrophilic additives of NVP [34].

We present a simple strategy for the fabrication of nitrogen-doped carbon using functionalized PVDF and polyvinylpyrrolidone (PVP) as a nitrogen source. Benefiting from the heteroatom doping, the NC-800 shows good electrochemical and dye adsorption performances.

## 2. Experimental Section

### 2.1. Materials and Characterization

PVDF (Solef) was obtained from Solvay Inc. (Brussels, Belgium) and was dried at 80 °C for 24 h before use. NVP (C_6_H_9_ON, AR, CAS No. 88-12-0, purified through vacuum distillation to remove the inhibitors before use) was purchased from Aladdin Chemistry Co. Ltd., and used as monomers to introduce N into the PVDF matrix. N-methyl-2-pyrrolidinone (NMP) (C_5_H_9_NO, AR, CAS No. 872-50-4) was purchased from Chengdu Kelong Inc. (Chengdu, China), and used as solvent. Azo-bis-isobutryonitrile (AIBN) (C_8_H_12_N_4_, AR, CAS No. 78-67-1) and N,N′-Methylenebisacrylamide (MBA) (C_7_H_10_N_2_O_2_, AR, CAS No. 110-26-9) were purchased from Aladdin Chemistry Co., Ltd. (Shanghai, China), and were used as initiator and cross-linking agent, respectively. Methylene blue and methyl orange were purchased from Aladdin Chemistry Co., Ltd. and used as probe dyes in this study. Deionized water was obtained from a pure water production system and used throughout the experiments. The morphologies of porous carbon were recorded with scanning electron microscopy (SEM) (Hitachi 4800, Tokyo, Japan) and transmission electron microscopy. X-ray photoelectron spectroscopy (XPS, ESCALab220i-XL, London, UK) was performed with monochromatic Al-Ka radiation (300 W) and Raman spectra (Renishaw, Wotton-under-Edge, UK) were recorded with excitation laser of 514 nm. Fourier transform infrared spectra (FTIR, Thermo Nicolet, Waltham, MA, USA) were obtained within the wavenumber range of 4000–800 cm^−1^ and thermogravimetric analysis (TGA, TG209F3, Berlin, Germany) was conducted in N_2_ atmosphere.

### 2.2. Synthesis of N-Containing Polymeric Precursor via In Situ Cross-Linked Polymerization

N-containing polymeric precursor solution was prepared through in situ cross-linked polymerization of N-vinylpyrrolidone (NVP) in PVDF solutions. Firstly, 12 wt.% of PVDF was added to NMP till it was dissolved completely at 60 °C, followed by adding the monomers of NVP, with the concentration of 5 wt.%. After the NVP was added to PVDF solution homogeneously, the solution was added to a mixture of MBA and AIBN. The resulting mixture was stirred at 80 °C for 8 h, resulting in cross-linked polymerization. The amounts of AIBN and MBA added to the monomers of NVP were 1 wt.% and 3 mol.%, respectively. The whole process was under nitrogen protection to remove the oxygen. Spin coating in combination with a phase inversion approach was used in this study to synthesize N-polymeric precursor membranes. After in situ cross-linked polymerization, the N-containing polymeric precursor solution was spin coated on glass surfaces, followed with immediate immersion into deionized water. To remove the residual solvent effect, the produced N-containing polymeric precursor solution membranes were rinsed several times with deionized water and stored in deionized water for two weeks before use.

### 2.3. Synthesis of Heteroatom-Doped Porous Carbon

The heteroatom-doped porous carbon material was prepared by directly annealing the obtained functionalized PVDF at 800 °C for 2 h under N_2_.

## 3. Results and Discussion

The surface functional groups of the PVDF and N-containing polymeric precursor membrane were analyzed by FTIR spectroscopy, as shown in Figure 1. Upon comparison with pristine PVDF, after modification (N-containing polymeric precursor), the typical peak was observed at approximately 1670 cm^−1^, representing the carbonyl group in the PVP chain. The FTIR spectrum demonstrated the successful immobilization of N in the PVDF matrix via in situ cross-linked polymerization of NVP in PVDF solutions.

To investigate the doping element in carbon materials, the surface element contents and chemical valences of NC-800 were analyzed using XPS. The XPS full spectrum exhibited the C and N content of NC-800 and C-800 (Figure 2A,C,D). The peak is located at 285.14, representing the C–N components. The peak located at 402.4 eV corresponds to pyrrole N. The element composition of N-doped carbon and carbon is listed in Table 1. The results indicate that doped N atoms may promote better carbon surface wetting in an aqueous electrolyte, thereby increasing the conductivity of carbon materials. The Raman spectrum (Figure 2B) further verified the degree of graphitization. The spectrum showed the characteristic peaks along with D-band at 1352 cm^−1^ and G-band at 1585 cm^−1^ which represent the defect of carbon lattice and graphitization degree of carbon, respectively. The I_D_/I_G_ value of C-800 (0.91) was significantly higher than that of NC-800 (0.78), indicating that C-800 contained a lot of surface defects. As shown in Figure 2E and Appendix A, After N-doping, the porous structure of NC-800 is conducive to electronic transmission and storage. The BET surface of NC-800 and C-800 is 115.6 m^2^/g and 9.2 m^2^/g, respectively (Figure 2F).

A three-electrode system was used to assess the electrochemical performances (the GCD, CV, EIS curves) of NC-800 and C-800. At 50 mV/s, the NC-800 CV curves (Figure 3A) exhibited excellent symmetry, indicating that it is an electric double-layer capacitor. Moreover, NC-800 had a larger enclosed area than C-800, indicating that it possessed a greater capacitance. The CV curves exhibit little deformation at high scanning rates, indicating the device’s excellent structural stability (Figure 3B). To determine the specific capacitance quantitatively, the GCD curve of NC-800 at 1 A/g was examined (Figure 3C). The results showed the specific capacitance of NC-800 is 205 F/g at 1 A/g, which is higher than that of C-800 materials (108 F/g); this is attributed to its unique structures and N atom doping. Figure 3D exhibits the GCD curves of NC-800 at 1–10 A/g, the capacitances were 205, 150, 125, 96, 80 F/g and their typical symmetric shapes revealing an ideal EDLCs.

In Figure 4A, NC-800 delivers higher specific capacity than that of C-800 at different current densities. The EIS was used to analyze the resistance and the curves of the NC-800 and C-800 materials were studied over a range of 0.01–100 kHz and the Nyquist plots were obtained (Figure 4B). The NC-800’s fitting diagram revealed that the electrode appeared to be nearly straight in the low-frequency region, indicating that the NC-800 had an excellent rate of ion diffusion The charge transfer resistance (Rct) of 0.04 Ω was obtained in the high-frequency region. At high frequency, the narrower the semicircle, and at low frequency, the straight line indicates that the electrode has excellent capacitance properties. The comparison of results showed that the impedance of C-800 was higher (2 Ω), which might be attributed to the low conductivity of carbon structures. The long-term cycling performance of NC-800 and C-800 were investigated. The capacity retention for NC-800 and C-800 was 95.2% and 86.8%, respectively, after 5000 cycles (Figure 4C).

To determine the N-doped porous carbon’s adsorption capacity in this study, the probe dyes MB and MO were employed, and the adsorption levels of the two dyes are shown in Figure 5. The figure reveals that the maximum adsorption amounts of MB and MO are approximately 39 mg and 40 mg, respectively. The adsorption ratios of these two dyes are 780 mg/g and 800 mg/g, respectively. The porous structures revealed by SEM images were attributed to the results of adsorption capacities of N-doped porous carbons. Furthermore, adsorption saturation was reached after 30 min, as shown in Figure 5. The data on the adsorption capabilities of N-doped porous carbons on MB and MO revealed that N-doped porous carbons may be employed as adsorption material in water purification.

This adsorption capacities of N-doped porous carbon for MB are comparable with similar carbon-based adsorbents, such as activated carbon, hydrophobic activated carbons functionalized by ethylamine [35], seaweed biochar [36], magnetized palm shell waste-based activated carbon [37], etc. The adsorption capacities of each carbon-based adsorbent for MB are 270, 393, 513, and 163 mg/g, respectively. This phenomenon is attributed to the existence of the elements of N and O in the prepared porous carbon materials, which affect the hydrogen bonding/electrostatic interactions between N-doped porous carbon adsorbents and cationic MB and MO molecules [38].

## 4. Conclusions

In summary, N-doped hierarchical porous carbon was fabricated by in situ cross-linked polymerization of NVP in PVDF solution, followed by carbonizing of functionalized PVDF. Owing to the synergistic effect of structure and N doping, the NC-800 electrode shows higher specific capacitance (205 F/g). Furthermore, the adsorption capacity of NC-800 of MB and MO was 780 mg/g and 800 mg/g, respectively. The good electrochemical and adsorption performances show that NC-800 is a promising material for energy storage and adsorption.

## Figures and Tables

**Figure 1 micromachines-13-01747-f001:**
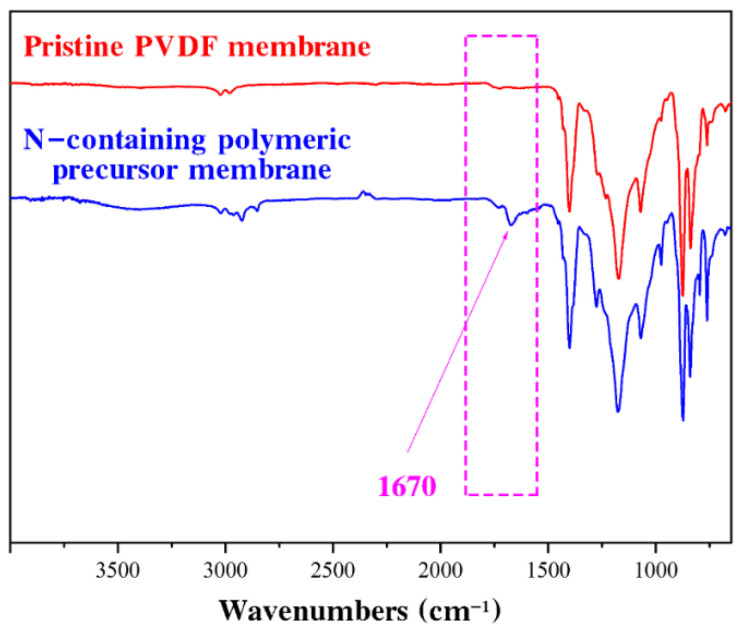
FTIR spectra of pristine PVDF and N-containing polymeric precursor membrane.

**Figure 2 micromachines-13-01747-f002:**
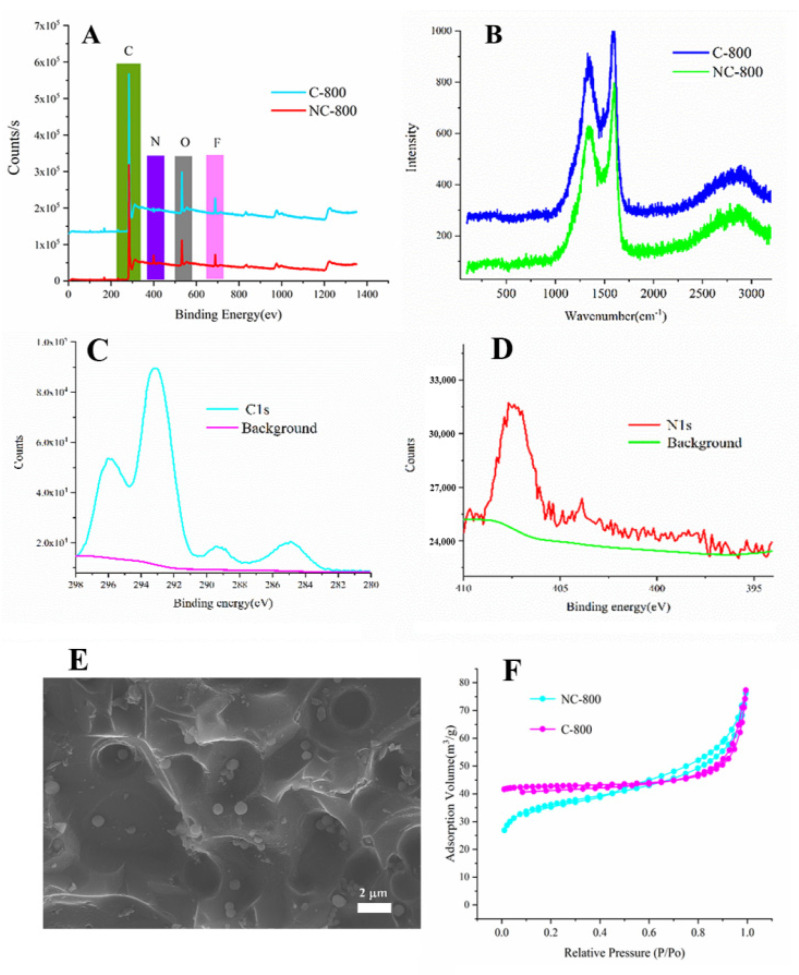
(**A**) XPS spectra of the C−800 and NC−800 samples; (**B**) C1S XPS spectra of the NC−800 samples; (**C**) N1S XPS spectra of the NC−800 samples; (**D**) Raman spectra of NC−800 and C−800; (**E**) SEM images of NC−800; (**F**) nitrogen sorption isotherms of NC−800 and C−800.

**Figure 3 micromachines-13-01747-f003:**
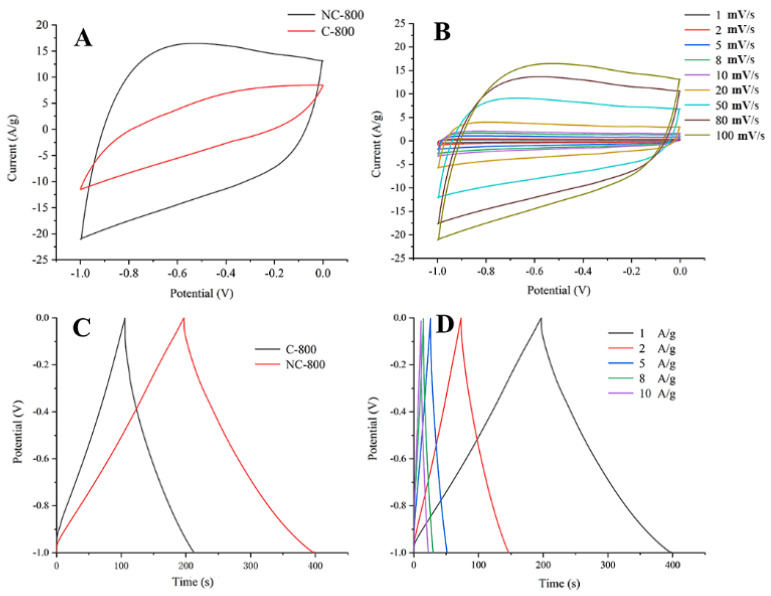
(**A**) CV curves of NC−800 and C−800 at 50 mV/s; (**B**) CV curves for NC−800 at 1−100 mV/s; (**C**) GCD curves of NC-800 and C-800 at 1 A/g; (**D**) GCD curves of NC-800 at 1−10 A/g.

**Figure 4 micromachines-13-01747-f004:**
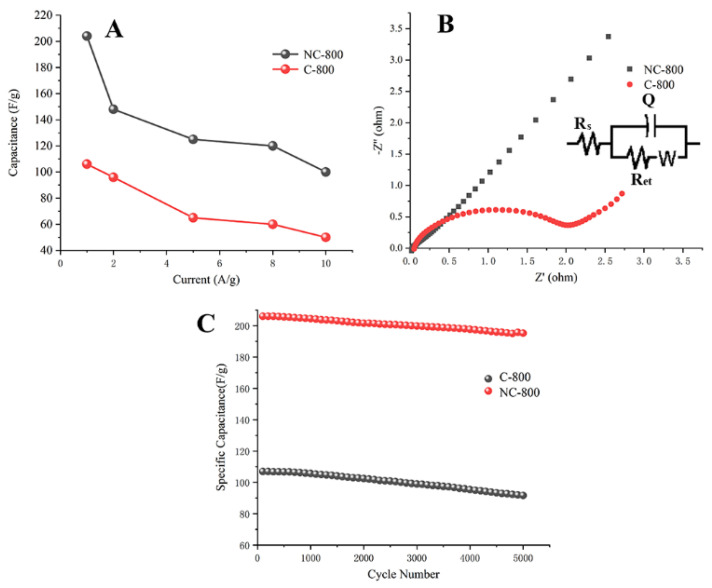
(**A**) Specific capacities of NC−800 and C−800 at different specific currents; (**B**) Nyquist plots of NC−800 and C−800; (**C**) cycle stability of NC−800 and C−800.

**Figure 5 micromachines-13-01747-f005:**
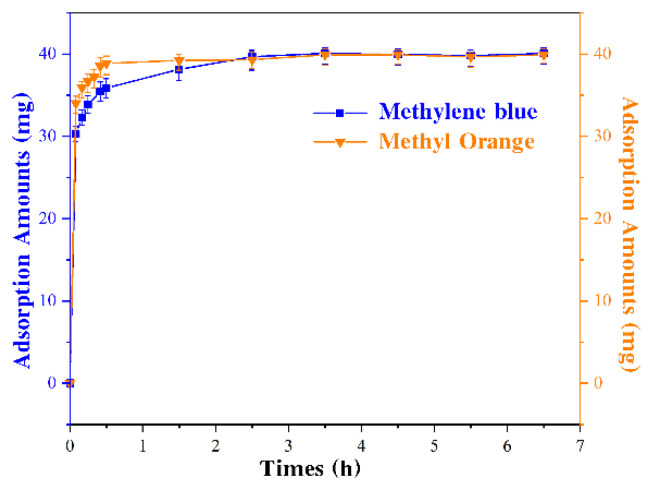
Adsorption capacity of N-doped porous carbons of methylene blue and methyl orange.

**Table 1 micromachines-13-01747-t001:** The elemental composition of carbon and N-doped carbon.

	C Atom%	O Atom%	N Atom%	F Atom%
N-doped Carbon	89.5%	5.8%	2.6%	3.1%
Carbon	92%	3.9%	1.5%	2.6%

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
