# Peer review of "Nitrogen Self-Doping Carbon Derived from Functionalized Poly(Vinylidene Fluoride) (PVDF) for Supercapacitor and Adsorption Application"

_micromachines, 2022, doi:10.3390/mi13101747_

Round 1

Reviewer 1 Report

The paper entitled “Nitrogen Self-doping Carbon Derived from Functionalized 2 Poly(vinylidene fluoride) (PVDF) for High-Performance Supercapacitor Electrodes” by Yi Wang et al. is focused on a strategy for the fabrication of nitrogen-doped carbon from functionalized PVDF and its use in supercapacitors and dyes absorption. The results in both applications are not so relevant but the novelty of fabrication process make this work worthy of consideration by the scientific community. The paper is globally well conceived, bibliography is sufficient, illustrations are clear and adequate. However, many aspects of the manuscript should be improved and the following points should be addressed:

1) Title. A specific capacitance of 205 F/g is relatively low if compared to other new methods to fabricate porous carbon (i.e. from biomass), therefore this value does not justify the “High-Performance” written in the title. Please remove it and mention the adsorption application.

2) Abstract. Please specify the current density (1 A/g) also for the specific capacitance. 

3) The entire text should be carefully checked because many imprecisions are present. Some examples: raw 36 repetition of porosity concept, raw 42 “enhance” and “increase” are wrong, raw 47 repetition of dislocations, raw 52 point in place of the semicolon, raw 99 “dissolved”, raw 186 “mattix”, raw 189 “suggest” is wrong.

4) Introduction. PVDF cannot be considered eco-friendly. Ecotoxicity assessment studies have been performed (e.g. Toxics 2022, 10, 479) and it is well known its processing in some harmful solvents (i.e. NMP in this work). Please remove the concepts of sustainability and eco-friendliness of PVDF or better explain these aspects.

5) Please explain the mechanism of self-doping

6) Since the authors present a new porous carbon in the literature, they should provide the surface area values (BET Analysis).

7) Please describe the structure of supercapacitors.

For these reasons, this paper is suitable for publication on Micromachines journal after major revision.

Author Response

Dear Editors and Reviewers.

Thank you for your letter and for the reviewers’ comments concerning our manuscript entitled " Nitrogen Self-doping Carbon Derived from Functionalized Poly(vinylidene fluoride) (PVDF) for Supercapacitor and adsorption application" (ID: micromachines-1952584). Those comments are all valuable and very helpful for revising and improving our paper, as well as the important guiding significance to our researches. We have studied comments carefully and have made correction which we hope meet with approval. Revised portion are marked in red in the paper. The main corrections in the paper and the responds to the reviewer’s comments are as following:

The paper entitled “Nitrogen Self-doping Carbon Derived from Functionalized 2 Poly(vinylidene fluoride) (PVDF) for High-Performance Supercapacitor Electrodes” by Yi Wang et al. is focused on a strategy for the fabrication of nitrogen-doped carbon from functionalized PVDF and its use in supercapacitors and dyes absorption. The results in both applications are not so relevant but the novelty of fabrication process make this work worthy of consideration by the scientific community. The paper is globally well conceived, bibliography is sufficient, illustrations are clear and adequate. However, many aspects of the manuscript should be improved and the following points should be addressed:

  • A specific capacitance of 205 F/g is relatively low if compared to other new methods to fabricate porous carbon (i.e. from biomass), therefore this value does not justify the “High-Performance” written in the title. Please remove it and mention the adsorption application.

Thanks for your suggestion. The high performance is removed and the adsorption application is added.

  • Please specify the current density (1 A/g) also for the specific capacitance.

Thanks for your suggestion. The current density is added.

  • The entire text should be carefully checked because many imprecisions are present. Some examples: raw 36 repetition of porosity concept, raw 42 “enhance” and “increase” are wrong, raw 47 repetition of dislocations, raw 52 point in place of the semicolon, raw 99 “dissolved”, raw 186 “mattix”, raw 189 “suggest” is wrong.

Thanks for your carefully checking.

The line 36: The high porosity is deleted.

The line 42: The enhance and increase is revised as improve.

The line 47: The dislocations is deleted.

The line 52: The semicolon is revised as point.

The line 99:The dissolved is corrected as added.

The line 186: The mattix is deleted.

The line 189: The suggest is revised as show

  • PVDF cannot be considered eco-friendly. Ecotoxicity assessment studies have been performed (e.g. Toxics 2022, 10, 479) and it is well known its processing in some harmful solvents (i.e. NMP in this work). Please remove the concepts of sustainability and eco-friendliness of PVDF or better explain these aspects.

Thanks for your advice. The sustainability and eco- friendliness are removed.

5) Please explain the mechanism of self-doping.

Thanks for your suggestion. Self doping is achieved by using heteroatoms existing in biomass or polymers itself, such as carbonized seaweed or kelp Self doping of nitrogen and oxygen can be realized( M. H. Wang, Z. Q. Xu, H. J. Du, Z. Guo, Y. F. Yang, J. W. Fu. One-step fabrication of porous carbon microspheres with in situ self-doped N, P, and O for the removal of anionic and cationic dyes. Diamond and Related Materials, 2022, 126, 109123).

  • Since the authors present a new porous carbon in the literature, they should provide the surface area values (BET Analysis).

Thank you for the advice. The BET Surface of NC-800 and C-800 is 115.6 m2/g and 9.2 m2/g, respectively(Figure 2d)

7) Please describe the structure of supercapacitors.

Thanks for your suggestion. In the three electrodes system: A platinum foil Hg/HgO are counter and reference electrode, respectively. The electrolyte is 6M KOH, mass loading of carbon and n-doped carbon are 4 mg respectively.

In the two electrodes system: The prepared carbon materials (mass ratio 80%), carbon black(mass ratio 10%) and PVDF(mass ratio 10%) was mixed and loaded on a nickel foam(1× 1 cm2) to fabricate the working electrode. Another prepared carbon electrode is used as counter and reference electrode. The electrolyte is 6M KOH, mass loading of N-doped carbon is 4 mg respectively.

Reviewer 2 Report

 Y. Wang et. al. have reported the synthesis of nitrogen-doped carbon derived from functionalized PVDF. The supercapacitor and absorption properties have been investigated. A few major issues need to resolve before the acceptance of the manuscript. 

1. Experimental part needs to be improved. How the authors carried electrochemical measurements, which electrolyte is used for measurement purposes? mass loading of carbon and n-doped is not provided.

2. Information about the current collector is missing.

3. The equations used for the calculation of specific capacitance and for absorption are missing.

4. SEM is not clear, the difference between the morphology of N-doped carbon and carbon should be provided.

5. Deconvolution of XPS spectra be better, more detailed analysis should be provided.

6. The values of Rct need to be recalculated. I think the values given are the x-axis intercept of the EIS spectra which is Rs, not Rct.

7. The equivalent circuit should be provided.

8. Authors should compare their supercapacitor as well as absorption studies with the previous literature. 

Author Response

Dear Editors and Reviewers.

Thank you for your letter and for the reviewers’ comments concerning our manuscript entitled " Nitrogen Self-doping Carbon Derived from Functionalized Poly(vinylidene fluoride) (PVDF) for Supercapacitor and adsorption application" (ID: micromachines-1952584). Those comments are all valuable and very helpful for revising and improving our paper, as well as the important guiding significance to our researches. We have studied comments carefully and have made correction which we hope meet with approval. Revised portion are marked in red in the paper. The main corrections in the paper and the responds to the reviewer’s comments are as following:

Reviewer 2

  1. Wang et. al. have reported the synthesis of nitrogen-doped carbon derived from functionalized PVDF. The supercapacitor and absorption properties have been investigated. A few major issues need to resolve before the acceptance of the manuscript.
  2. Experimental part needs to be improved. How the authors carried electrochemical measurements, which electrolyte is used for measurement purposes? mass loading of carbon and n-doped is not provided.

Thanks for your suggestion. The electrolyte is 6M KOH, mass loading of carbon and n-doped carbon are 4 mg respectively. The information is added in the supporting information.

  1. Information about the current collector is missing.

Thanks for your suggestion. The current collector is Ni foam. The information is added.

  1. The equations used for the calculation of specific capacitance and for absorption are missing.

Thanks for your suggestion. The equations used for the calculation of specific capacitance and for absorption are added in the supporting information.The values of capacitance were deduced from through followed equations.

C (F/g) =I×Δt/mΔV                                 (1)

where I (A), Δt (s), m (g) and ΔV (V) on behalf of the discharge current, time for discharge, mass of the carbon material and voltage window, respectively.

For the supercapacitor device, the energy density E (Wh kg -1) and power density P (W kg -1) were calculated based on the following equation:

E = 0.5 × Cs ×∆V2 /3.6

P =3600 E /∆t.

Where Cs (F g-1), ∆V (V) ∆t (s) denote the specific capacitance, the potential change, the discharging time.

  1. SEM is not clear, the difference between the morphology of N-doped carbon and carbon should be provided.

Thanks for your suggestion. The SEM images of N-doped carbon and carbon are provided.

The SEM image of carbon is added:

The SEM image of N-doped carbon is added:

  1. Deconvolution of XPS spectra be better, more detailed analysis should be provided.

Thanks for your suggestion. The C and N XPS spectra is added.

  1. The values of Rct need to be recalculated. I think the values given are the x-axis intercept of the EIS spectra which is Rs, not Rct.

Thanks for your advice. Rct (evaluated from the semicircle intercept) of N-doped carbon is 0.07 Ω, Rct of carbon is 2 Ω.

  1. The equivalent circuit should be provided.

Thanks for your advice. The equivalent circuit is provided.

  1. Authors should compare their supercapacitor as well as absorption studies with the previous literature.

This adsorption capacities of N-doped porous carbon for MB are comparable with the similar carbon-based adsorbents, such as activated carbon, hydrophobic activated carbons functioned by ethylamine [35], seaweed biochar [36], magnetized palm shell-waste based activated carbon [37] and so on. The adsorption capacities of each carbon-based adsorbent for MB are 270, 393, 513 and 163 mg/g, respectively. This phenomenon is attributed to the existence of the elements of N and O in the prepared porous carbon materials, which affect the hydrogen bonding/electrostatic interactions between N-doped porous carbon adsorbents and cationic MB and MO molecules [38].

References:

[35] E.I. El-ShafeySyeda, N.F. Ali, S. Al-Busafi, Preparation and characterization of surface functionalized activated carbons from date palm leaflets and application for methylene blue removal, J. Environ. Chem. Eng. 4 (2016) 2713–2724.

[36] M.J. Ahmed, P.U. Okoye, E.H. Hummadi, B.H. Hameed, High-performance porous biochar from the pyrolysis of natural and renewable seaweed (Gelidiella acerosa) and its application for the adsorption of methylene blue, Bioresour. Technol. 278 (2019) 159–164.

[37] K.T. Wong, N.C. Eu, S. Ibrahim, H. Kim, Y. Yoon, M. Jang, Recyclable magnetite-loaded palm shell-waste based activated carbon for the effective removal of methylene blue from aqueous solution, J. Clean. Prod. 115 (2016) 337–342.

[38] Z. Heidarinejad, O. Rahmanian, M. Fazlzadeh, M. Heidari, Enhancement of methylene blue adsorption onto activated carbon prepared from date press cake by low frequency ultrasound. J. Mol. Liq. 264 (2018) 591–599.

Reviewer 3 Report

The reviewer believes that if the manuscript is considered for publication, it should be accepted after the following comments are taken care of:

Q1: P1L39 Authors mentioned that “…carbon could not offer higher capacitance due to the poor conductivity.” The sentence is confusing. Researchers often use carbon materials to increase conductivity of the base materials.

Q2: P1L63 The sentence should be “… polymerization and cross-linking occur…”.

Q3: P1L68 Include full definition of “NVP”.

Q4: P2L88 “(Hitachi 4800…) end bracket is missing.

Q5: P2L92-93 It should be “cm-1” and “N2”. Please correct this throughout the manuscript.

Q6: P3L120 Include full definition of “PES”.

Q7: P3 Present the elemental composition from the XPS survey scan.

Q8: P4F2 Include SEM image of C-800 to show the reported difference between C-800 and NC-800.

Q9: P4L140 It should be “mV/s”. Same goes for the Figure 3B.

Q10: P5L156 It should be “kHz”.

Author Response

Dear Editors and Reviewers.

Thank you for your letter and for the reviewers’ comments concerning our manuscript entitled " Nitrogen Self-doping Carbon Derived from Functionalized Poly(vinylidene fluoride) (PVDF) for Supercapacitor and adsorption application" (ID: micromachines-1952584). Those comments are all valuable and very helpful for revising and improving our paper, as well as the important guiding significance to our researches. We have studied comments carefully and have made correction which we hope meet with approval. Revised portion are marked in red in the paper. The main corrections in the paper and the responds to the reviewer’s comments are as following:

The reviewer believes that if the manuscript is considered for publication, it should be accepted after the following comments are taken care of:

Q1: P1L39 Authors mentioned that “…carbon could not offer higher capacitance due to the poor conductivity.” The sentence is confusing. Researchers often use carbon materials to increase conductivity of the base materials.

Thanks for your suggestion. The sentence is revised as Relatively high cost, and scarce raw material resources hinder the large-scale production of these carbons and their application in the supercapacitor.

Q2: P1L63 The sentence should be “… polymerization and cross-linking occur…”.

Thanks for your suggestion. The sentence is revised as“… polymerization and cross-linking occur…”.

Q3: P1L68 Include full definition of “NVP”.

Thanks for your carefully checking. N-vinyl pyrrolidone is short for NVP. N-vinyl pyrrolidone is added in the manuscript.

Q4: P2L88 “(Hitachi 4800…) end bracket is missing.

Thanks for your carefully checking. The bracket is added.

Q5: P2L92-93 It should be “cm-1” and “N2”. Please correct this throughout the manuscript.

Thanks

Q6: P3L120 Include full definition of “PES”.

Thanks for your advice. full definition of “PES” is Polyethersulfone.

Q7: P3 Present the elemental composition from the XPS survey scan.

Table 1 The elemental composition of carbon and N-doped carbon

C atom%

O atom%

N atom%

F atom%

N-doped Carbon

89.5%

5.8%

2.6%

3.1%

Carbon

92%

3.9%

1.5%

2.6%

Q8: P4F2 Include SEM image of C-800 to show the reported difference between C-800 and NC-800.

Thanks for your suggestion. The SEM images of C-800 and NC-800 are added.

The SEM image of carbon is added:

The SEM image of N-doped carbon is added:

Q9: P4L140 It should be “mV/s”. Same goes for the Figure 3B.

Thanks for your carefully checking. All the mv/s is revised as mV/s.

Q10: P5L156 It should be “kHz”

Thanks for your suggestion, it revised as kHz.

Round 2

Reviewer 1 Report

The authors have properly made the suggested corrections and added the requested measurements. The revised manuscript is now suitable for the publication in Micromachines journal.

Reviewer 2 Report

The authors have resolved all my queries. The present version of the manuscript can be accepted for publication.